# Keep on Swimming: Real Attackers Only Need Partial Knowledge of a Multi-Model System

**Julian Collado** *
HiddenLayer Inc.
jcollado@hiddenlayer.com

**Kevin Stangl**
HiddenLayer Inc.
kstangl@hiddenlayer.com

## Abstract

Recent approaches in machine learning often solve a task using a composition of multiple models or agentic architectures. When targeting a composed system with adversarial attacks, it might not be computationally or informationally feasible to train an end-to-end proxy model or a proxy model for every component of the system. We introduce a method to craft an adversarial attack against the overall multi-model system when we only have a proxy model for the final black-box model, and when the transformation applied by the initial models can make the adversarial perturbations ineffective. Current methods handle this by applying many copies of the first model/transformation to an input and then re-use a standard adversarial attack by averaging gradients, or learning a proxy model for both stages. To our knowledge, this is the first attack specifically designed for this threat model and our method has a substantially higher attack success rate (80% vs 25%) and contains 9.4% smaller perturbations (MSE) compared to prior state-of-the-art methods. Our experiments focus on a supervised image pipeline, but we are confident the attack will generalize to other multi-model settings [e.g. a mix of open/closed source foundation models], or agentic systems.

## 1 Introduction

Recent AI research has shown the effectiveness of agentic architectures and systems composed of multiple models that decompose problems and create scaffolds in a solution pipeline[16, 29, 38, 27, 36, 6]. Alternatively consider an initial model doing a complex pre-processing step for a second model, for example a foundation model[8, 10, 1, 2, 34, 18, 37] that processes the input and passes its output to another model for a classification or some other task. In production systems, a service is often a pipeline of multiple pre/post-processing steps based on heuristics and machine learning models. Combining models this way has proven to be very effective and will likely increase over time with the rise of multi-agent systems.

The proliferation of real world AI systems and the horizon of ever more powerful methods has made securing these models against malicious or un-authorized use ever-more urgent. Model providers have responded to these security threats by implementing a mix of a) including safety fine-tuning [19] b) weaker side-car models that halt the model from responding based on detecting malicious queries or harmful outputs [35] c) closed model weights to prevent an attacker from developing white-box attacks [31] d) rate-limiting the users of a centrally served model to avoid black-box attacks [15].

However, the conmingling of multiple models, of closed/open source, introduces new security vulnerabilities that are not precisely captured by existing threat models and complicates defense based on keeping the weights hidden or rate limiting the user to avoid the creation of proxy models.

---

*Primary and corresponding Author

38th Conference on Neural Information Processing Systems (NeurIPS 2024).

*We will show how to attack a system of models even when an adversary has restricted access to part of this system such that they cannot create a proxy for the first models/components of the system.*

White-Box attacks [22] assume perfect knowledge of the model weights, allowing gradient based optimization techniques to find adversarial perturbations. Black-Box attacks [14, 32, 23] achieve a similar effect to the White-Box attacks but without having access to the weights, instead the attackers can only query the model with different inputs but may have varying degrees or knowledge about the model architecture, biases and other parameters. Black-box attacks either typically train a proxy model[32] or estimate local gradients to find perturbations for specific inputs. Grey-Box attacks are similar to Black-Box attacks; one Grey-Box model could consist of White-Box access to part of a system and Black-Box access to another component. For example in an encoder-decoder architecture, the attacker may have White-Box access to the encoder and Black-Box access to the decoder.

## 1.1   Threat Model: Multi-Model System Attack With Partial Proxy Access

We introduce a new and realistic threat model for multi-agentic and multi-model applications that we first test in a vision modality.

In the simplest case, consider a system that is a composition of two models, e.g. $h_1$ and $h_2$, so the overall output is $\hat{y} = h_2(h_1(x))$. Specifically, we have black-box access to both models but it is only feasible[2] to create a proxy model for $h_2$ as shown in Figure 1. The proxy model for $h_2$ allows us to perform gradient based attacks against $h_2$, so we can compute a $\delta_{adv}$ such that $h_2(x_{mod} + \delta_{adv}) \neq y_{pred}$ where $x_{mod} = h_1(x)$ and $y_{pred}$ is the predicted label of $x$. In the rest of the paper, we refer to $x_{mod}$ as the output of model $h_1$. *The key difficulty in this scenario is that the transformation applied by $h_1$ might destroy the adversarial modification such that $h_1(x + \delta_{adv}) \neq x_{mod} + \delta_{adv}$ and therefore $h_2(h_1(x + \delta_{adv})) = y_{pred}$.*

*We focus on the case when the modifications applied by the $h_1$ are reversible* in the sense that $x_{mod}$, and $x_{mod} + \delta_{adv}$, can be "re-inserted" into $x$. Consider the case where $h_1$ is a segmentation model that detects a region of interest and crops the image and we have designed an adversarial perturbation attacking the cropped subset of the full image. That adversarial perturbation could be re-inserted into the original image inside the crop box. Formally, $h_1 : \mathcal{X} \rightarrow \mathcal{X}$ and $h_2 : \mathcal{X} \rightarrow \mathcal{Y}$, for some input modality $\mathbb{X}$. This allows us to "re-insert" the adversarial sample $x_{mod} + \delta_{adv}$ crafted for $h_2$ into $x$ to create an adversarial sample for the whole system. Another example of a pair of models that satisfies this property could be a pair of natural language models; where the first model processes a piece of text, generating a new text string, that is then handed off to the second model for the final computation.

## 1.2   Our Contributions

To our knowledge, we propose the first attack specifically for a multi-model compositional problem where a proxy is only available for the last model. We observe that this is a more realistic scenario for industrial applications where it might not be feasible to create a proxy for each section of the system or where adversary might not have access to information about the first sections of the system, for example the pre-processing of the data or an adversarial defense, but the last leg of the system might be approximated with an open source model or have been trained in a public dataset.

We provide an iterative method, which we name the *Keep on Swimming Attack* (*KoS*, pronounced chaos) to ensure that the attack survives the modifications applied by the non-proxy-able sections of the system, and show our attack has a higher success rate and lower noise levels than the natural baseline method, based on Expectation over Transformation (EoT) [4]. In Appendix A.1, we show how an end-to-end black-box attack was ineffective in this setting; it is this dead end that motivated us to design and develop the *KoS* Algorithm. *Our method shows that even if a system has a secure and restricted section, there are instances in which the overall system can still be exploited with adversarial attacks.*

---

[2]Due to limited computational or query budget for the first model while the second model is more accesible or has an open weights version.

Figure 1: Multi-Model System with Gradient Restrictions: We have limited query access to $h_1$ and full query/gradient access to $h_2$ and want to craft an end-to-end attack. The core issue is that the adversarial sample against $h_2$ (second row) might not remain adversarial after the transformation of $h_1$. E.g. in the case where $h_1$ is a segmentation and image crop, the perturbation could slightly modify the crop box out of $h_1$, such that the sample is no longer adversarial to $h_2$ (third row).

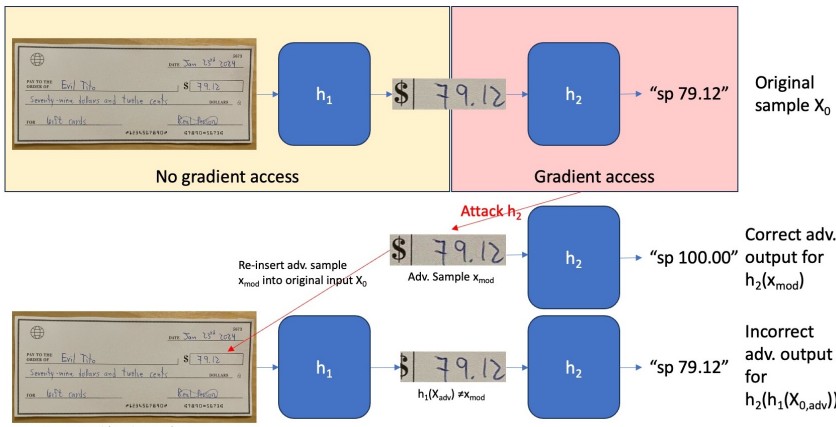

## 2   Related Work

Our setting is similar to the Expectation over Transformation [4] method when the first model $h_1$ is thought of as an arbitrary transformation instead of a learned model. In that work, the transformations are physically motivated and represent parametric transformations of the input like lighting and camera noise. In general, the attacker must know enough about the first transformation to sample from the family of transformations, which is different from our threat model, where we only have query access to the first model. This allows the creation of a set of transformation input points, to be used for averaging gradients. This is the primary competing method and we conduct baseline experiments using this method.

BPDA (Backward-Pass Differentiable Approximation)[3], designed to attack systems that intentionally obfuscate gradients for security reasons, uses a differentiable proxy model to craft gradient based attacks. It is challenging to apply this method in our setting, since in contrast to the defenses attacked in [3], creating a good proxy for a full-size model is a meaningful task and our paper *focuses on the case when creating such a proxy is not feasible, e.g. rate-limiting defense or attacker resource constraints like information, computation, and query limits* [3].

HopSkipJumpAttack[13] could be used for end-to-end black-box attacks in a system like the one we propose since it does not require a proxy for $h_1$. However, in our experiments we found that while this attack was able to achieve the desired target, the adversarial noise introduced was too large to be considered a successful human adversarial attack (see Appendix A.1).

There has been previous work that considers multi-model systems, for example treating the modifications applied by optics and image processing pipelines in cameras as $h_1$ and a classification model as $h_2$ [33]. However, this attack creates a proxy model for $h_1$ which is not possible in our problem.

Recent work [24] has shown adversaries can compose multiple-'safe' models to achieve 'unsafe' behavior and prior work in algorithmic fairness and strategic classification, [7, 21, 20, 9, 17] showed that even in the context of supervised binary classification the composition of 'fair' models can result in highly 'unfair' outcomes. Our work suggests a similar effect is present in adversarial robustness; having a 'safe' (e.g. black-box, non-proxy-able) section of the system does not guarantee the safety of the overall system.

Very intriguingly [12], extracted exact information from a production grade language model, e..g the exact projection embedding layer, in a top-down manner meaning the algorithm extracted information

---

[3]Future work will characterize the query complexity of *KoS*. From our experiments we expect *KoS* will show a substantial decrease in query complexity compared to creating a proxy model.

Figure 2: Keep on Swimming (*KoS*) Multi-Model Attack: Update the sample fed into the start of the pipeline whenever the adversarial perturbation is made ineffective by $h_1$

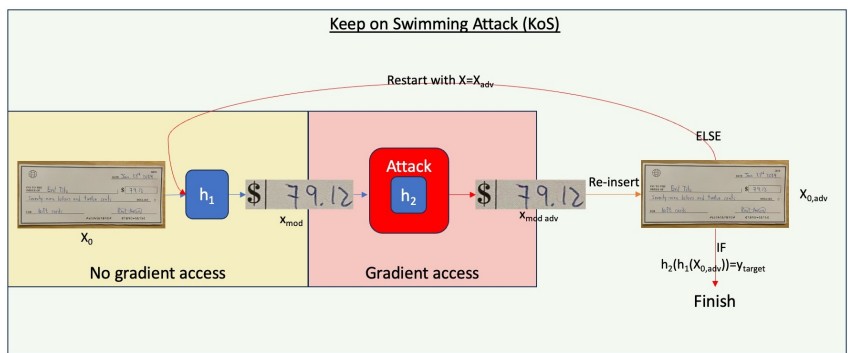

from the final layers of the neural network. Demonstrated vulnerabilities like this, combined with our algorithm, could allow attackers to execute an effective end-to-end attacks on closed weight production grade systems using their partial knowledge.

# 3   Method

We can easily craft gradient based attacks for $h_2$ using well known methods[22, 11, 30, 28] if we have white-box access to $h_2$ or have created a reliable proxy model. However, since we only have black-box access to $h_1$ and cannot train a proxy model for that component, we cannot directly craft an end-to-end gradient based adversarial attack $h_2(h_1(x))$. Furthermore since the modifications applied by $h_1$ are specific to each sample and thus each adversarial sample iteration, there is no guarantee that adversarial modifications against $h_2$ will survive the transformation applied by $h_1$.

*We propose an iterative method, the Keep on Swimming Attack. Simply update the sample that we will attack for $h_2$ when the adversarial perturbation has been removed by $h_1$, using the new output of $h_1$.*

Formally, attack $h_2$ and after $K$ gradient based attack iterations, re-insert the adversarial perturbation attacking $h_2$ into the original input and pass it through $h_1$ to check if the attack is still adversarial. If the adversarial perturbation survived the transformation of $h_1$, e.g. $h_1(x + \delta_{adv})$ is still in the same domain of $x_{mod}$, which in our experiment means whether the cropping box coordinates are unchanged, and if we have reached our goal e.g. $h_2(h_1(x + \delta_{adv})) = y_{target}$, we terminate and have achieved our objective of an end-to-end attack.

Else if the adversarial transformation survived $h_1$ but has not yet reached the adversarial target, e.g. $h_1(x + \delta_{adv}) = x_{mod} + \delta_{adv}$ and $h_2(h_1(x + \delta_{adv})) \neq y_{target}$ then we attack for $K$ more iterations.

Else if the adversarial sample was transformed/warped by $h_1$ and we have a new $x_{mod}$, so $h_1(x + \delta_{adv}) = x_{mod2}$, we just Keep on Swimming; we replace the adversarial sample that we had so far, $x_{mod} + \delta_{adv}$ with the new modified output of $x_{mod2}$ and keep attacking.

The attack finishes after a maximum number of iterations or when the end-to-end attack is successful. The algorithm is described in detail in Algorithm 1 and shown in Figure 2.

In Algorithm 1, the $ReInsert(x, x_{adv})$ operator takes the accumulated adversarial perturbations that have been applied to $x_{adv}$ and pastes it back into the original $x$. In our experiment this means pasting $x_{mod} + \delta_{adv}$ into the region of $x$ from which we extracted $x_{mod}$. While our proposed attack pipeline uses a gradient based attack against $h_2$, the pipeline is still valid for non-gradient based attacks.

While our experiments focus on this specific modality, we believe in the general applicability of our framework and Algorithm 1. One example of an application could be a system that processes and answers questions about a text. A first non-proxy-able model extracts quotes from the text related to the question and the second proxy-able model generates an answer. Our method makes is suitable for agentic architectures and in general systems where there is a sequential combination of either models or heuristics in which we only have a partial information.

---

**Algorithm 1** Keep on Swimming (*KoS*) Attack

---

$x_0, x_{mod}, y_{target}$ ▷ Original input, Output of $h_1$ and input of $h_2$, Target output for attack
$h_1(x_0) \rightarrow x_{mod}, h_2(x_{mod}) \rightarrow y_{pred}$ ▷ $h_1$ and $h_2$
$Attack(x_{mod}, y_{target}, h_2)$ ▷ Attack iteration on proxy-able section
$ReInsert(x_0, x_{mod})$ ▷ Function to re-insert adversarial modifications from $x_{mod}$ into $x_0$
$SameDomain(x_{mod.adv}, x_{mod}) \rightarrow bool$ ▷ Checks if values have the same domain
MaxRestarts ▷ Max number of restarts due to a different $x_{mod}$ domain
MaxIterations ▷ Max number times $K$ of attack iterations on a single $x_{mod}$. This also controls how frequently to obtain feedback from the $h_1$ transformation while crafting $\delta_{adv}$

$\delta_{adv} \leftarrow 0$
$x_{0,adv} \leftarrow x_0 + \delta_{adv}$ ▷ Initialize intermediate solution to $x_0$
$i \leftarrow 0$
**while** $i <$MaxRestarts **do**
    $x_{mod} \leftarrow h_1(x_{0,adv})$ ▷ reference for original domain
    $x_{adv} \leftarrow h_1(x_{0,adv})$
    $j \leftarrow 0$
    **while** $SameDomain(h_1(x_{0,adv}), x_{mod})$ and $j <$MaxIterations **do** ▷ Keep on Swimming
        **if** $h_2(h_1(x_{0,adv})) == y_{target}$ **then**
            Finish and return $x_{0,adv}$
        **end if**
        **for** k=1:K **do**
            $\delta_{adv} \leftarrow Attack(x_{adv}, y_{target}, h_2)$
            $x_{adv} \leftarrow x_{adv} + \delta_{adv}$
            $j \leftarrow j + 1$
        **end for**
        $x_{0,adv} \leftarrow ReInsert(x_0, x_{adv})$
    **end while**
    $i \leftarrow i + 1$
**end while**
return AttackFailure
    ▷ Note: $SameDomain(h_1(x_{0,adv}), x_{mod})$ checks that $h_1$ has not changed the domain of $x_{mod.adv} = h_1(x_{0,adv})$ from the original domain of $x_{mod}$ such that it destroys the attack. If the domain has changed we restart to adapt to the new domain.

---

## 4 Experiments

In order to simulate the scenario proposed in this paper we focus on the problem of creating an adversarial attack to cause the numerical value of a check to be misread. The input for this system is an image of a check. The first model of the system ($h_1$) consists of a segmentation model that identifies the areas of the image with text. The output of model $h_1$ is the area of the image containing the check's numerical amount ($x_{mod}$), written in latin numerals [4]. The second model of the system ($h_2$) is an OCR (optical character recognition) system that identifies the numerals in the image. To simulate the target system we use the CRAFT [5] segmenter ($h_1$) to create cropped one line text image. To obtain the text in each image ($h_2$), we used the publicly available Microsoft's Transformer based OCR for handwritten text[26]. We ran our experiments on a database of pictures of checks filled with handwritten information in which CRAFT was able to correctly identify and extract the numerical amount of the check. The attack objective is to transform the predicted numerical amount of one check to another value for a total of 20 attack samples.

For the attack, we assume black-box access to $h_1$ but not the possibility of creating a proxy. To create the adversarial sample for the image-to-text (OCR) section ($h_2$) we use the "I See Dead People" (ISDP)[25]. This attack is grey-box since it has white-box access to the image encoder but not to the text decoder. In this case we had white-box access to the image encoder since we used the same OCR model as CRAFT but a proxy model for the image encoder could have been used making this attack entirely black-box. Note that this does not affect the results since ISDP was used with all

---

[4]I.e. we only attack the numerical OCR part of the check and not the written text and latin numerals.

Table 1: Comparison of adversarial attack pipelines using the "I See Dead People" (ISDP) Image2Text attack. All values are averages that consider successful and failed attack attempts. Success rate is the percentage of attacks where the full check image output matches the target output. L-Full is the Levenshtein distance between the target output and the output when passing the full check image, $h_2(h1(x))$. L-Crop is the Levenshtein distance between the target output and the output of $h_2$ using the cropped image as input, $h_2(x)$ attack). MSE is the mean squared error on the full check image. Time is the average time to run the attack per sample in seconds.

| Method | Success Rate | L-Full | L-Crop | MSE | Time (s) |
|---|---|---|---|---|---|
| Original Image | 0% | 0 | 0 | 0 | 0 |
| ISDP Baseline[25] | 5% | 4.8 | 0.7 | **0.39** | **49.99** |
| Crop Robust ISDP[4] | 25% | 2.25 | 0.85 | 0.53 | 375.29 |
| Keep on Swimming ISDP | **80%** | **1.1** | **0.05** | 0.48 | 85.09 |

attack pipelines and the objective of the *KoS* attack pipeline is to create an adversarial sample that survives $h_1$ and is still effective for $h_2$. The *KoS* attack is not affected by how the adversarial sample was created for $h_2$.

### 4.1 Benchmarks

We compare our method with a baseline of only attacking $h_2$ and re-inserting the adversarial cropped image into the original large image (ISDP Baseline). We also compare our method with creating attacks that are robust to the transformation from $h_1$ using the method from [4] (Crop Robust ISDP). For the Crop Robust ISDP, we take a slightly larger crop than the one from the starting image, perform 10 random crops such that the text is always contained in the crop, attack each random crop independently, average the gradients and update the image to create the adversarial version. We found these hyperparameters to provide the best overall results for this method.

We compare the methods in terms of the attack success rate, the mean squared error (MSE) with respect to the original image, and computational cost. Table 1 shows the success rate of the *KoS* pipeline is considerably higher and the Levenshtein distance the final output of both the cropped and the full image are considerably lower than just using the ISDP attack or creating a version that is robust to cropping.

The *KoS* pipeline introduces more noise and takes more time than the Baseline ISDP but less than the Crop Robust ISDP attack. The key benefit of our method, that clearly Pareto dominates the other methods is our substantially higher attack success rate. We would note that we investigated these alternate baseline methods first and it was only our inability to craft successful attacks that required us to design the *KoS* attack.

## 5 Conclusion

We have shown how adversaries can use their knowledge of one model in a multi-model system to craft effective end-to-end attacks with the *KoS* algorithm. Further work is needed to study the convergence properties of *KoS*, and generalizing the attack to other settings like attacking a composition of LLMs. That said, these initial results already demonstrate the need for timely research into attacks and defenses in the threat model of Multi-Model Systems With Partial Proxy Access. If multi-agent and multi-model systems inherit the vulnerability of the most 'proxy-able' model, that suggests serious un-patched vulnerabilities already exist in the foundation model era, and we can expect the impact of such vulnerabilities to be amplified in the upcoming era of agentic AI.

## Acknowledgments and Disclosure of Funding

We are grateful to HiddenLayer for supporting this research and its publication.

# A Appendix

## A.1 HopSkipJumpAttack

We attempted to use the HopSkipJumpAttack on the system but failed to produce samples where the attack is adversarial for human viewers, i.e. perturbations do not change the true label. Figure 3 shows one a sample where the initial number 25.86 is misread as the target output 100.00.

Figure 3: Adversarial attack sample using HopSkipJumpAttack; the adversarial modification is too evident to be useful.

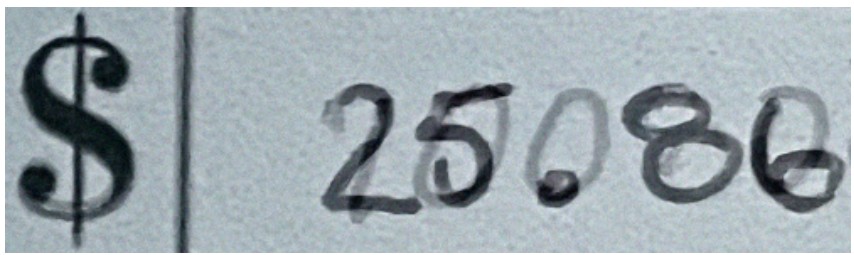

# B Visual Comparison of Adversarial Samples

Figure 4: Visual comparison of final cropped images for each attack pipeline converting 79.12 value to 100.00 and vice-versa showing if the attack was successful or not. The final adversarial sample is the whole check image but here we show the cropped versions to highlight visual differences on the adversarial modifications. One can observe the *KoS* samples have less noticeable perturbations in this particular sample as reflected by the lower average MSE from Table 1.

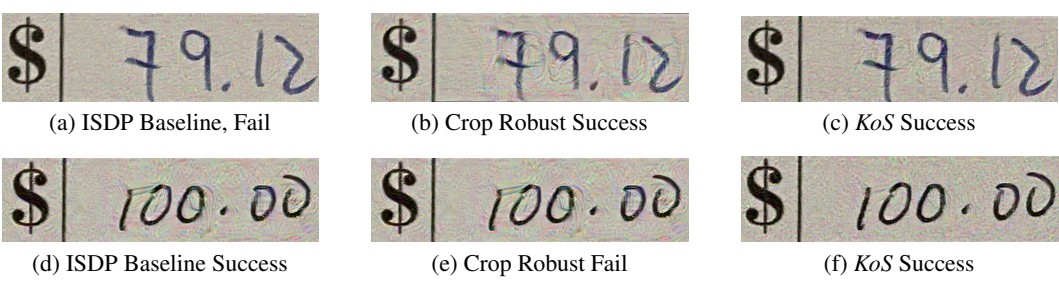

(a) ISDP Baseline, Fail      (b) Crop Robust Success      (c) *KoS* Success

(d) ISDP Baseline Success      (e) Crop Robust Fail      (f) *KoS* Success

## C   Social Impact Statement

Our paper takes an adversarial approach to disclose possible vulnerabilites for systems of machine learning models; we demonstrate a new attack on composed models. Using the attack would require a new attacker to obtain knowledge about the attacked system.

Unlike papers that publish jailbreaks or zero-days, our disclosure cannot be used immediately off the shelf to attack production grade systems. That said, we are currently working on a generalization of this work that could be used to target systems currently in production.

This attack is very natural and well-motivated so it is possible or even likely similar attacks exist in-the-wild and are being used by real world attackers, so we believe introducing and studying the vulnerability in this proof-of-concept will allow for the design and deployment of effective defenses to this vulnerability.

One interpretation of our work, *which we do not advocate for*, is that releasing model weights could allow for attackers to break real world multi-model systems. Thus securing the modern AI stack requires locking down model weights. This would be an inversion of the well known Kerchoff's Principle from cryptography. We note, but do not advocate, for this interpretation which would no doubt have a significant social impact even though it is difficult to forecast if it would be positive or negative.

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
