# OpenReview forum: "Keep on Swimming: Real Attackers Only Need Partial Knowledge of a Multi-Model System"
_NeurIPS.cc/2024/Workshop/SafeGenAi — SafeGenAi Poster_

### Official Review · Reviewer_8T9k · 2024-10-09
**The paper introduces an attack method for multi-model systems with partial access, showing some empirical results, but it lacks theoretical analysis, broader evaluations, and clarity in some sections.**

**Rating:** 3
**Confidence:** 4

**Review:**

### Summary:

This paper introduces an adversarial attack method, *Keep on Swimming* (KoS) for multi-model systems where an attacker has limited access. The study focuses on a two-model pipeline, where the adversary can only create a proxy for the second model, while the first model’s transformations may disrupt adversarial perturbations.

### Pros:

1. **High Attack Success Rates**: Demonstrates an improvement in success rate compared to prior state-of-the-art methods.

### Cons:

1. **Experiment**:
- The experiment is restricted to a single dataset.
- The sample size is only 20 examples.
- Some details about the experiments are unclear, e.g. models used, the hyperparameters, and the exact setup.
- The paper does not convincingly demonstrate that this attack can be applied on real-world models
- There is no exploration of how this method performs against existing adversarial defense techniques.
2. **Incomplete Explanation of Attack on h2**: This paper does not provide details regarding how the user obtains the h2 (proxy) model and how to generate adversarial example on it.
3. **Theoretical Analysis**: The paper does not provide formal theoretical analysis for the KoS algorithm.
4. **Novelty**: The paper primarily combines existing adversarial attack methods and model extraction techniques

### Lint:
There are minor grammatical errors in the paper (e.g., "a system that takes takes in a text and questions about it").

---

### Official Review · Reviewer_vLKF · 2024-10-09

**Rating:** 8
**Confidence:** 4

**Review:**

### **Summary**
The authors propose a novel adversarial attack method that outperforms state-of-the-art (SOTA) techniques by achieving higher attack success rates with smaller perturbations in a multi-modal system. This approach is effective even when attackers only have access to a proxy model for the final black-box system($h_2$), and when input adversarial perturbations may be disrupted due to initial model($h_1$) transformations.

### **Pros**
- The approach is novel and utilizes a simple, brute-force-like algorithm to reach the target within a predefined iteration limit.
- The method is clearly articulated and easy to understand.

### **Cons**
- The current method involves a gray-box attack on $h_2$, with the authors suggesting its extension to cases where only a proxy estimate of $h_2$ is available. However, this can potentially lead to unexpected divergences of $delta_{adv} \leftarrow Attack(x_{adv}, y_{target}, h_2)$, wasting attack iterations thus reducing attack success rate.

### **Questions**
- The authors claim that their method results in **9.4% smaller perturbations** than prior SOTA approaches, but they provide no quantitative evidence beyond the statement about a visual *One can observe KoS samples have less noticeable perturbations*. It would be beneficial to include more quantitative data to substantiate this claim.
- The approach is relevant to image cropping and classification/recognition tasks, but further exploration into its application across other multi-modal systems would be interesting.

---

### Official Review · Reviewer_uYCD · 2024-10-10
**Good paper discussing original method for applying adversarial attacks on multi-modal systems**

**Rating:** 7
**Confidence:** 3

**Review:**

The paper mainly deals with attacking a multi-modal model hat has two subcomponents: a segmentation model to identify important parts of the image, then an image-to-text model that processes the segments. The proposed method of adversarial attacks (KoS) continuously attacks the second component, *h2* until the desired target output is reached or *maxIterations* have been reached. Here is a list of pros and cons from reading the paper:

**Pros**:
- Method is very reproducible, as the authors have listed every dataset and models used in the experiments.
- Experiments show positive results using their method and include examples of successes/fails for clarity
- Detailed algorithmic description of the method that is easy to follow

**Cons**:
- Little discussion of how this method would apply to other model types, such as composition of several LLMs or other single-modality systems.
- This method is far slower than baseline methods since it is iterative; there is little discussion regarding this drawback

---

### Official Review · Reviewer_i3jL · 2024-10-10
**Multi-Model Attack with Limited Proxy Access: Promising Results, Room for Expansion**

**Rating:** 6
**Confidence:** 3

**Review:**

This paper introduces the "Keep on Swimming" (KoS) attack, a novel approach for targeting multi-model systems where an attacker has only partial proxy access. The authors present a realistic threat model and demonstrate the effectiveness of their method compared to baseline approaches. Algorithm 1 makes the KoS algorithm easily reproducible. However, the paper could benefit from generalizing the approach beyond just h1 and h2 to a system of h1, h2, h3, ..., hi models.

The experiments, while demonstrating the potential of the KoS attack, are limited in scope. The dataset of 20 samples seems small for drawing robust conclusions. Additionally, it's not explicitly stated whether these are targeted or untargeted attacks. The analysis seems untargeted, but it should be clarified. Despite these limitations, the results show a substantial improvement in attack success rate (80% vs 25%) compared to previous methods, with smaller perturbations. I also appreciate the computational run times presented.

The paper is well-structured and the algorithm is clearly presented, making the approach easy to understand.
A minor suggestion would be to relabel the "Keep on Swimming ISDP" method in Table 1 as "KoS ISDP (Ours)" to make the authors' contribution more immediately apparent to readers.

Overall, this work highlights an important vulnerability in multi-model systems and opens up avenues for future research on both attacks and defenses in this domain.